# Symbol-shift Equivariant Neural Networks

## Abstract

Neural networks have been shown to have poor compositionality abilities: while they can produce sophisticated output given sufficient data, they perform patchy generalization and fail to generalize to new symbols (e.g. switching a name in a sentence by a less frequent one or one not seen yet). In this paper, we define a class of models whose outputs are equivariant to entity permutations (an analog being convolution networks whose outputs are invariant through translation) without requiring to specify or detect entities in a pre-processing step. We then show how two question-answering models can be made robust to entity permutation using a novel differentiable hybrid semantic-symbolic representation. The benefits of this approach are demonstrated on a set of synthetic NLP tasks where sample complexity and generalization are significantly improved even allowing models to generalize to words that are never seen in the training set. When using only 1K training examples for bAbi, we obtain a test error of 1.8% and fail only one task while the best results reported so far obtained an error of 9.9% and failed 7 tasks.

## 1 Introduction

Previous work have shown how neural networks fail to generalize to new symbols (Lake & Baroni, 2018; Sinha et al., 2019; Hupkes et al., 2019). In particular, Lake & Baroni (2018) showed that seq2seq models are able to perfectly learn a set of rules given enough data, however they fail to generalize these learned rules to new symbols.

We illustrate the generalization issue of current models in the context of question-answering (QA) on the first task of bAbi (Weston et al., 2015). This dataset identified a set of tasks testing which type of reasoning can be achieved by a question-answering system (e.g. several supporting facts, compound reference, positional reasoning, etc). Each task consists in a set of stories with an associated question such as:

> "John took the apple. John traveled to the hallway.
> Who has the apple?"

Clearly, we would expect a QA system to be able to answer the previous example when "John" is replaced by "Sasha", "Bob", or any possible name even if it has not been seen during training. To investigate whether QA models perform abstraction over symbols, we perform an experiment where the training and test sets of the first bAbi task are regenerated with an increasing number of names. Fig. 1 shows how the performance of Memory-Networks (MN) (Sukhbaatar et al., 2015) and Third-order

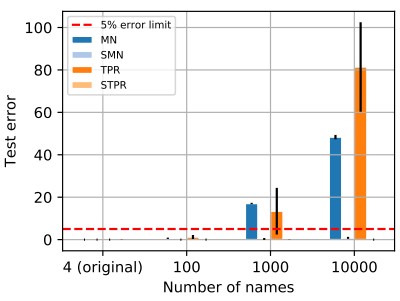

Figure 1: Test error on first bAbi task when increasing the number of names. Error of symbolic models are all bellow 1%.

tensor product RNN (TPR) (Schlag & Schmidhuber, 2018) dramatically drops as the number of names increases in contrast to their symbolic counter-part SMN and STPR proposed in this paper. Both models reach low error well below the 5% limit even when the number of names and vocabulary become considerably larger than the original task.

The main contribution of this paper is the proposal of a hybrid semantic/symbolic representation that is equivariant to entity permutation. The main advantage and novelty of our approach is that entities are not required to be identified in advance as we rely solely on differentiation to determine whether a word acts like an entity. We show how to extend two question-answering models to handle this hybrid representation and demonstrate in extensive experiments the benefit of such an approach: the sample-complexity is significantly improved, better compositionality is obtained and symbolic models reach better accuracy on the studied tasks in particular when being trained with less training data.

The paper starts by reviewing related works, we then introduce formally what it means to permute entities. We then define layers that are robust to such perturbation and show how two recent question-answering models can be adapted in this context. Finally, experiments are conducted to assess the benefits of our method.

## 2 RELATED WORKS

Improving compositionality of neural networks has been an important on-going effort in the past years. The SCAN dataset proposed from Lake & Baroni (2018) initially showed how standard neural networks baselines can fail to generalize to new symbols when learning a set of artificially constructed rules. Several approaches were proposed to solve this issue. For instance, Lake (2019) designed meta-learning episodes that led the model to solve the task, (Nye et al., 2020) showed how one could infer symbolic neural programs with a similar meta-learning procedure. Alternatively, Gordon et al. (2020) proposed to design an equivariant model (a model whose latent representations are unchanged when permuting symbols). A common limit of those approaches is that they require specifying which words are symbols in advance (Lake (2019); Nye et al. (2020) also requires substantial amount of supervision and designing meta-episodes). An exception is Russin et al. (2019) which proposed to decompose syntax and semantic for SCAN. None of those approaches can generalize to arbitrary large amount of entities or entities not seen in the training as the one we propose.

The problem of compositionality becomes much easier if symbols (or entities) are detected beforehand. For instance, Li et al. (2015) showed that replacing entities by dedicated token placeholders leads to significant improvement in question answering. The same approach has also been applied in Machine Translation and Data to Text generation Luong et al. (2015); Serban et al. (2016); Lebret et al. (2016), to enable sequence to sequence models to generalize to unseen words at inference time. While specifying entities in advance (or detecting them in a pre-processing step with Named-entity recognition (Marsh & Perzanowski, 1998)) before applying a model may give compositionality, we would clearly want instead models to be able to infer automatically whether a word should behave as a symbol or not. While positional encoding (Graves et al., 2014; Vaswani et al., 2017) may give some compositionality - as it allows to reason over positions - this solution is not practical for language as inter-word distances are not fixed. For instance the distance between a noun and its verb varies and positional embedding is not enough to achieve compositionality (Hupkes et al., 2019).

An interesting line of research have been the study of equivariant models whose representations are invariant (or equivariant) to symmetries present in the data (Zaheer et al., 2017; Ravanbakhsh et al., 2017). Adding invariance to data symmetries has been theoretically shown to drastically reduces sample complexity (Sannai & Imaizumi, 2019). For instance, convolution neural networks require significantly less training data and achieve much better performance than a MLP as they are invariant to image translation. Gordon et al. (2020) proposed the first NLP model provably capable of handling symmetries between symbols albeit requiring the need to specify such symmetries in advance.

Tensor product representation (TPR) Smolensky (1990) allows to stores complex relations between value and variables with distributed representations and offer some compositionality. Recently, Schlag & Schmidhuber (2018) proposed an architecture able to learn TPR parameters by differentiation and obtained state-of-the-art results for bAbi at the time of publishing. However, the compositionality of the proposed approach is limited (as shown in Fig. 1) by the fact that every entity needs to be seen sufficiently many times so that a proper entity vector is found, in addition the model has been shown to learn orthogonal representation for entities which requires as many hidden dimensions as the total number of entities.

Figure 2: Illustration of symbol-shift equivariance. Left: representation of word parameters (identical words are depicted close so that they can be distinguished). Middle and right: two cases where $\Phi$ is symbol-shift equivariant for two different symbol-shifts.

Finally, the VARS approach (Vankov & Bowers, 2020) consists in outputting a one-hot vector representing a symbolic variable that is randomly assigned to different positions during training to enforce compositionality. While this approach grants some compositionality, the approach is limited as one must draw symbol permutations so that each object is seen in all possible one-hot values. In addition, one must specify in advance which object or word behaves as a symbol and the method only support symbolic output and cannot represent symbolic inputs nor perform computation with hybrid representation as we propose (combining semantic and symbolic representation).

## 3   SYMBOL-SHIFT EQUIVARIANCE

When we learn to answer "John" given a specific context we would like to be able to answer "Sasha" if both names were permuted in the context. In what follows, we introduce the notion of symbol-shift equivariance: e.g. a condition restricting possible permutations as some permutations may perturb the sentence grammar (permuting "John" by "why") or cause ambiguity (permuting "John" by "Mary" if the question involves a gender).

We assume all words are taken from a vocabulary $V$ which is a discrete set of $n$ words. We are interested in providing an answer $a \in V$ given a context consisting in a question $q = [q_1, \ldots, q_{n_q}] \in V^{n_q}$ and a list of sentences (or stories) $x = [x_1, \ldots, x_T]$ with $x_i = [x_{i1}, \ldots, x_{in_i}] \in V^{n_i}$. We denote $X = (x, q)$ and $\Phi(X) = a$ the function that predicts the answer $a \in V$ given the context $X = (x, q) \in \mathcal{X}$.

Let $\Gamma : V \to V$ a word permutation. Given a sequence $[y_1, \ldots, y_n]$, we define $\Gamma(y) = [\Gamma(y_1), \ldots, \Gamma(y_n)]$ where the permutation is applied to each word in the sequence, similarly, $\Gamma(X) = ([\Gamma(x_1), \ldots, \Gamma(x_T)], \Gamma(q))$.

Assuming each word has an associated vector parameter, we say that a permutation is a *symbol-shift* if it preserves vectors parameters. For instance in Figure 2, a map permuting "John" with "Sasha" is a symbol-shift as both words share the same parameters but a map permuting "John" and "lemon" is not. Formally, assuming each word $i \leq n$ has an associated set of parameters $v_i \in \mathbb{R}^D$, we say that a permutation $\Gamma : V \to V$ is a symbol-shift if $v_i = v_{\Gamma(i)}$ for all $i \leq n$.

We are now ready to define symbol-shift equivariance. A critical advantage is that we do not need to specify symmetries in advance between entities, as we instead rely on vector semantics whose embeddings will be learned end-to-end.

**Definition 1.** *Let $\Phi : \mathcal{X} \to V$ a function mapping a context to an answer. We say that $\Phi$ is symbol-shift equivariant if for any symbol-shift $\Gamma$ and for any $X \in \mathcal{X}$,*

$$\Phi(\Gamma(X)) = \Gamma(\Phi(X))$$

## 4   SYMBOLIC QUESTION ANSWERING

In this section, we show how to define symbol-shift-equivariant models. The main idea consists of concatenating two representations, a standard semantic representation in $\mathbb{R}^d$ as well as a symbolic representation in $\mathbb{R}^m$ where $m$ denotes the number of distinct words in the stories and question. The symbolic representation will be made such that for $i \leq m$, the $i$-th component of the symbolic representation corresponds to the $i$-th word appearing in the context. For instance in Figure 3, there are $m = 5$ words in the context and "apple" is the fourth word by order of appearance so its symbolic

embedding is the fourth one-hot vector. The model symbolic output has larger probability for the fourth word of the context which is dereferenced to "Apple".

We now describe formally how the symbolic representations are constructed and how we perform linear transformation and projection back to the original vocabulary. Finally, we derive the symbolic counter-part of Memory-Networks and TPR models that will be proved to be symbol-shift equivariant.

**Mapping words into and from symbolic representations.**  In each input example, the set of words present in the stories $x$ and the question $q$ is denoted as

$$C_X = \{q\} \cup \{\{x_i\}, i \leq T\}$$

and we let $m = |C_X|$ be the number of distinct words in the context. To project words to its symbolic representation $\mathbb{R}^m$, we represent each unique word by a one-hot vector representing the position of its first appearance, using the bijection $\varphi_X : C_X \to [1, m]^1$.

To dereference a symbolic representation in $\mathbb{R}^m$ back to its vocabulary id, we define the matrix $B_\varphi \in \mathbb{R}^{n \times m}$ that maps one-hot vectors of symbolic representations to one-hot vectors in $\mathbb{R}^n$ representing the word id in the vocabulary as shown in Fig. 3, such that $B_\varphi e_j^m = e_{\varphi^{-1}(j)}^n$ for $j \leq m$ where $e_l^k \in \mathbb{R}^k$ $^2$ denotes the $l$-th one-hot vector in $\mathbb{R}^k$.
Note given a one-hot symbolic representation $\tilde{p} \in \mathbb{R}^m$, the $i$-th coordinates of $B_\varphi \tilde{p} \in \mathbb{R}^n$ is given by :

$$[B_\varphi \tilde{p}]_i = \begin{cases} \tilde{p}_{\varphi(i)} & \text{if } i \in C \\ 0 & \text{else} \end{cases} \tag{1}$$

hence $B_\varphi$ allows to dereference a symbolic representation $\tilde{p} \in \mathbb{R}^m$ to a word vector $B_\varphi \tilde{p} \in \mathbb{R}^n$.

**Hybrid semantic-symbolic embeddings.**  We embed words with the concatenation of a standard semantic word embedding as well as a symbolic embedding respectively parametrized by $A \in \mathbb{R}^{d \times n}$ and $\alpha \in [0, 1]^n$. The *semantic* embedding maps a word $x \in [1, n]$ to $Ae_x \in \mathbb{R}^d$. While the *symbolic* embedding of $x$ consists in the one-hot vector of the order of appearance of the word $x$ in its context multiplied by a learned parameter. More precisely, it is defined as $\alpha_x e_{\varphi(x)} \in \mathbb{R}^m$, where $\alpha_x$ is an output of a sigmoid unit on a learnable parameter, i.e. $0 < \alpha_x < 1$, that indicates how much each word should behave as a symbol $^3$, and $e_{\varphi(x)}$ is the one-hot vector of the order of appearance of the word $x$ in its context. The final embedding of a word of $x$ then consists in the concatenation of the semantic and symbolic parts:

$$x \mapsto Ae_x \oplus \alpha_x e_{\varphi(x)} \in \mathbb{R}^{d+m}, \tag{2}$$

where $\oplus$ denotes the concatenation operator. Note that all parameters are differentiable allowing the model to learn both word semantic and how much each word should behave as a symbol. The symbolic part will be shown to allow the model to be robust to symbol permutation while being able to represent an arbitrary number of symbols and generalize to new ones. For instance in Fig. 3, one can see that permuting "John" by "Sasha" would not change embeddings (as long as both name share the same parameters) as both words will still appear in the same order in the context.

**Symbolic projection.**  Given an internal state $h = h_{\text{sem}} \oplus h_{\text{sym}} \in \mathbb{R}^{d+m}$, we interpret it as a distribution on the vocabulary $p \in \mathbb{R}^n$ with

$$p_{\text{sem}} = \text{softmax}\,(Bh_{\text{sem}}) \quad p_{\text{sym}} = B_\varphi \text{softmax}\,(h_{\text{sym}}) \tag{3}$$

$$p = \beta\, p_{\text{sem}} + (1 - \beta)\, p_{\text{sym}} \tag{4}$$

where $B \in \mathbb{R}^{n \times d}$ and $\beta \in [0, 1]$ are parameters to learn.

The final distribution is a mixture of two distributions $p_{\text{sem}}$ and $p_{\text{sym}}$. The first one $p_{\text{sem}}$ is seen as a semantic output as it increases answer probability of a word if its semantic embedding is closer

---

[1] From now on, we drop the subscript notation when there is no ambiguity and denote $C, \varphi$

[2] we later omit superscript dimension as they will always be implicitly defined

[3] Note that if $\alpha_x = 0$ for all word $x$, the model is reduced to a standard semantic-only model

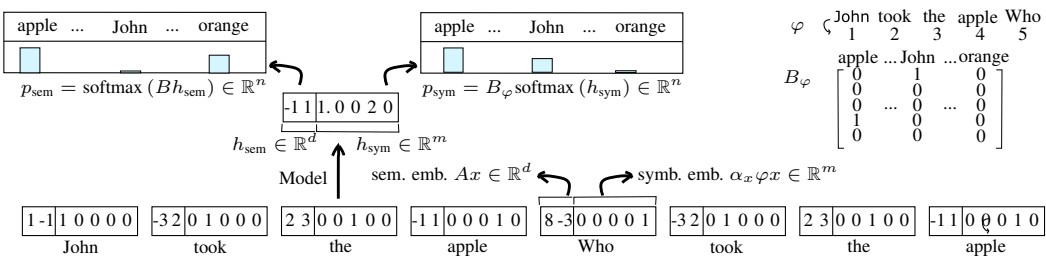

Figure 3: Illustration of symbolic representation in a case where $d = 2$ and $m = 5$.

to the semantic state $h_{\text{sem}}$. The second one $p_{\text{sym}}$ interprets $h_{\text{sym}}$ as probabilities of words from the context using $B_\varphi$ to dereference positions. Indeed, denote $\tilde{p} = \text{softmax}(h_{\text{sym}}) \in \mathbb{R}^m$ and using Eq. 1, we get that the $i$-th coefficient of $p_{\text{sym}}$ is given by

$$[p_{\text{sym}}]_i = [B_\varphi \tilde{p}]_i = \begin{cases} \tilde{p}_{\varphi(i)} & \text{if } i \in C \\ 0 & \text{else} \end{cases} \tag{5}$$

**Symbolic transformation.** We perform a linear transformation of an internal symbolic representation $h = h_{\text{sem}} \oplus h_{\text{sym}} \in \mathbb{R}^{d+m}$ with:

$$h_{\text{sem}} \oplus h_{\text{sym}} \mapsto W h_{\text{sem}} \oplus (\lambda I + \gamma \mathbb{1}\mathbb{1}^T) h_{\text{sym}} \in \mathbb{R}^{d+m}, \tag{6}$$

where $W \in \mathbb{R}^{d \times d}$, $\lambda \in \mathbb{R}$, $\gamma \in \mathbb{R}$ are parameters to learn, $I \in \mathbb{R}^{m \times m}$ is the identity matrix and $\mathbb{1} = [1, \ldots, 1]^T \in \mathbb{R}^{m \times 1}$. The linear transformation for the symbolic part is taken from Zaheer et al. (2017) where it was shown to be the unique form of a linear parametric equivariant function. In our case, the symbolic transformation is invariant to permutation of symbolic coordinates, allowing the model to be independent from the choice of a particular bijection $\varphi$.

## 4.1 Symbolic models

We are now ready to show how question-answering models can be extended so that they become symbol-shift equivariant. We recall the definition of Memory-Networks (Sukhbaatar et al., 2015) and Third-order tensor product RNN (Schlag & Schmidhuber, 2018) models before deriving their symbolic counter-part.

**Memory-Networks.** Memory-Networks iteratively updates an internal question representation with $K \geq 1$ self-attention layers (or *hops*) before projecting the final representation to the answer distribution. The model parameters consists in $K + 1$ embedding matrices $A^{(0)}, \ldots, A^{(K)} \in \mathbb{R}^{d \times n}$. The query representation is initially set to $u^{(0)} = \sum_{j=1}^{n_q} A^{(0)} e_{q_j}$ and iteratively updated with:

$$m_i = \sum_{j=1}^{n_i} A^{(k)} e_{x_{ij}}, \quad c_i = \sum_{j=1}^{n_i} A^{(k+1)} e_{x_{ij}} \tag{7}$$

$$u^{(k+1)} = u^{(k)} + \sum_{i=1}^{n} \text{softmax}\left(m_i^T u_i^{(k)}\right) c_i \tag{8}$$

In words, the internal representation is updated with the vector of output memories $c_i$ weighted by similarity between the current question representation $u_i^{(k)}$ and the input memory $m_i$. The final internal representation $u^{(K)} \in \mathbb{R}^d$ is mapped to the answer distribution with:

$$p = \text{softmax}\left(A^{(K)^T} u^{(K)}\right) \in \mathbb{R}^n$$

so that the probability of every word being the answer is a function of the similarity of its embedding in $A^{(K)}$ and the final question representation $u^{(K)}$. In addition the paper proposes temporal and positional encoding to allow distinguishing words or stories order which we discuss in the appendix.

**Symbolic Memory-Networks.** We extend Memory-Networks by using $K + 1$ symbolic embeddings from Eq. 2 and concatenating semantic and symbolic representation. This model has the parameters $A^{(k)} \in \mathbb{R}^{n \times d}$ and $\alpha^{(k)} \in \mathbb{R}^n$ for $0 \le k \le K$ as well as $\beta \in [0, 1]$. Instead of mapping internal representations into $\mathbb{R}^d$, we map them to $\mathbb{R}^{d+m}$ with:

$$m_i = \sum_{j=1}^{n_i} A^{(k)} e_{x_{ij}} \oplus \alpha_{x_{ij}}^{(k)} e_{\varphi(x_{ij})}, \quad c_i = \sum_{j=1}^{n_i} A^{(k+1)} x_{ij} \oplus \alpha_{x_{ij}}^{(k+1)} e_{\varphi(x_{ij})} \tag{9}$$

The internal representation is updated $K$ times with Eq. 8 to obtain a final internal representation $u^{(K)} \in \mathbb{R}^{d+m}$. To obtain the output predictive distribution, we project $u^{(K)} \in \mathbb{R}^{d+m}$ as described in Eq. 4 with $B^T = A^{(K)}$ and $\beta \in [0, 1]$.

**TPR.** Third-order tensor product RNN proposes to encode stories with a non-standard RNN whose representation is a tensor product representation (Schlag & Schmidhuber, 2018). More precisely, the model state consists in a tensor denoted as $F_t \in \mathbb{R}^{E \times R \times E}$ that is initialized with zeros and updated at each story with:

$$F_t = F_{t-1} + \Delta F_t$$

The update term $\Delta F_t$ depends on a learned parametric representation of entities and roles that are obtained by mapping the story representation $s_t$ with MLPs. The story representation is obtained by summing the $k$ words of a story with $s_t = \sum_{j=1}^{n_t} A e_{x_{tj}} \odot p_j$ where $\odot$ denotes the component-wise product, $A$ denotes an embedding matrix and $p_i$ are positional embeddings. Once the story representation is obtained, entities and roles representations are obtained with MLPs:

$$e_t^{(i)} = f_{e^{(i)}}(s_t; \theta_{e^{(i)}}) \in \mathbb{R}^E \quad r_t^{(j)} = f_{r^{(j)}}(s_t; \theta_{r^{(j)}}) \in \mathbb{R}^R \tag{10}$$

for $1 \le i < 3$, $1 \le j < 4$ and where $f$ is an MLP and $\theta$ its parameters. The update term $\Delta F_t$ is given by a close form formula designed to update entity information into $F_t$ given the entity and role embeddings $e_t^{(i)}$ and $r_t^{(i)}$, we detail it in the appendix for space reasons.

The internal representation $F_t$ is updated after reading each story and to perform inference, the final internal representation $F_T$ is used to decode the entity and role representation of the question. Similarly to the story embedding, entities and role representations of the questions are first obtained by mapping the question embedding $s_Q = \sum_{j=1}^{n_q} A e_{q_j} \odot p_j$ through a MLP:

$$n = f_n(s_Q; \theta_n) \in \mathbb{R}^R, \quad l_j = f_{l_j}(s_Q; \theta_{l_j}) \in \mathbb{R}^E, \quad 1 \le j < 4 \tag{11}$$

Given those representations, the distribution over possible answer is obtained by first obtaining the following internal representations:

$$\hat{i}^{(1)} = (F_T \bullet n) \bullet_{34} l^{(1)}, \quad \hat{i}^{(2)} = (F_T \bullet \hat{i}^{(1)}) \bullet_{34} l^{(2)}, \quad \hat{i}^{(3)} = (F_T \bullet \hat{i}^{(2)}) \bullet_{34} l^{(3)}, \tag{12}$$

where $\bullet_{34}$ denotes tensor inner-product, finally the answer distributions is obtained with the following projection:

$$p = \text{softmax}\left(B \sum_{k=1}^{3} \text{LN}(\hat{i}^{(k)})\right) \in \mathbb{R}^n \tag{13}$$

where LN denotes layer-normalization (Ba et al., 2015) and $B \in \mathbb{R}^{n \times E}$ are projection parameters. For MLPs, Schlag & Schmidhuber (2018) proposes to use two hidden layers with internal dimension $d$ and tanh activation functions, hyperparameters of the method are given in the appendix. The parameters of the models are word and positional embeddings, MLPs parameters as well as projections parameters.

**Symbolic TPR.** We modify TPR to handle symbolic representations for entity, role as well as intermediate representations. All entities, roles and intermediate representations are embedded into $\mathbb{R}^{d+m}$. Stories are embedded symbolically with:

$$s_t = \sum_{j=1}^{n_t} (A e_{x_{tj}} \odot p_j) \oplus (\alpha_{x_{tj}} e_{\varphi(x_{tj})}) \in \mathbb{R}^{d+m}$$

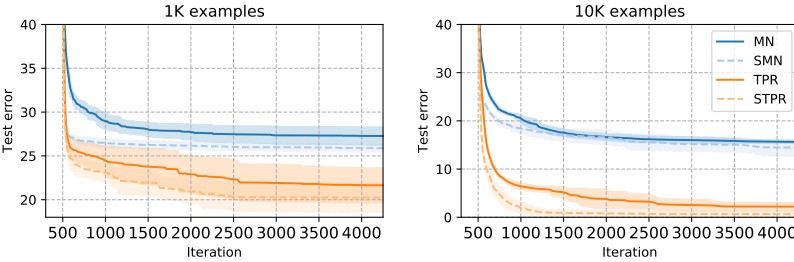

Figure 4: Learning curves of average accuracy when training with 1K and 10K examples of bAbi.

Then, we use symbolic MLPs to find entity and role representations:

$$e_t^{(i)} = \tilde{f}_{e^{(i)}}(s_t; \theta_{e^{(i)}}) \in \mathbb{R}^{d+m}, \quad r_t^{(j)} = \tilde{f}_{r^{(j)}}(s_t; \theta_{r^{(j)}}) \in \mathbb{R}^{d+m}, \tag{14}$$

where $\tilde{f}$ denotes a symbolic MLP with linear transformation as described in Eq. 6.

The tensor representation is updated with the same equations as TPR to obtain a final representation $F_T \in \mathbb{R}^{(d+m)^3}$. Given the symbolic embedding of the question $s_Q = \sum_{j=1}^{n_q}(Ae_{q_j} \odot p_j) \oplus (\alpha_{q_j} e_{\varphi(q_j)})$, we extract symbolic roles and entities with a symbolic MLP:

$$n = \tilde{f}_n(s_Q; \theta_n) \in \mathbb{R}^{d+m}, \quad l_j = \tilde{f}_{l_j}(s_Q; \theta_{l_j}) \in \mathbb{R}^{d+m} \tag{15}$$

We then obtain the hidden representation $h = \sum_{k=1}^{3} \text{LN}(\hat{i}^{(k)}) \in \mathbb{R}^{d+m}$ with Eq. 12 that is projected to an answer distribution as described in Eq. 4 where $B \in \mathbb{R}^{n \times d}, \beta \in [0, 1]$ are additional projection parameters.

A key result justifying the *symbolic* qualifier is that SMN and STPR are symbol-shift equivariant.

**Theorem.** *SMN and STPR are symbol-shift equivariant.*

The proof is in the appendix A for space reasons. The main idea consists in remarking that embeddings are invariant to symbol-shift and consequently, latent representations are invariant too since the model is deterministic. The proof then shows that having identical latents gives equivariance given our symbolic projection construction.

We observe that in practice, the learned semantic embeddings of different entities will not be exactly the same as in Figure 2: in particular the semantic embeddings of "apple" and "orange" may be close but different for instance. Our experiments will show that having symbol-shift equivariant models is a good inductive bias improving accuracy and compositionality of the studied models even if entities vectors are not exactly aligned.

## 5 EXPERIMENTS

We perform experiments on bAbi tasks (Weston et al., 2015) with version v1.2 of the dataset. The initial dataset consists in a set of 20 synthetic question-answering tasks designed to test capabilities of a dialog agent. In each example, an answer must be provided given a question and stories consisting in a sequence of sentences. We use the two versions which contains 1K and 10K training examples per task respectively. Given our computing budget, we perform all experiments in the *single-task* setting where models are trained on the 20 tasks independently. Every experiment is repeated with 10 different seeds and we report mean/std over those runs.

The performance of MN, SMN, TPR and STPR is reported in Fig. 4 and Tab. 1. Fig. 4 depicts the average error per task over time and shows that symbolic approaches significantly improves the sample efficiency of both methods which achieve much faster convergence rate and also converges to better values. In Tab. 1, we report the average error obtained at convergence for all methods. Again, symbolic models SMN and STPR outperforms their non-symbolic counter-part.

| original test-set | zero-shot room test-set | zero-shot object test-set | zero-shot people test-set |
|---|---|---|---|
| John took the apple in the kitchen.
John went to the bedroom.
Where is the apple? (bedroom) | John took the apple in the **kitchenette**.
John went to the **guest-room**.
Where is the apple? (**guest-room**) | John took the **key** in the kitchen
John went to the bedroom.
Where is the **key**? (bedroom) | **Sasha** took the apple in the kitchen.
**Sasha** went to the bedroom.
Where is the apple? (bedroom) |

Figure 5: Illustration of zero-shot test-datasets

Table 2: Test error when training on original bAbi tasks and evaluating error on different test-dataset. For zero-shot test-datasets, test entities (e.g. rooms, objects, people) are not seen during training.

| task | 2 | | | | 5 | | | | |
|---|---|---|---|---|---|---|---|---|---|
| dataset | original | room | object | people | original | room | object | people | average |
| MN | 21.8 ±0.4 | 99.5 ±0.6 | 38.2 ±1.3 | 67.1 ±2.8 | 14.8 ±2.7 | 14.9 ±1.3 | 48.7 ±1.4 | 24.6 ±0.8 | 41.2 |
| SMN (ours) | 3.4 ±0.6 | 100.0 ±0.0 | 62.7 ±2.3 | 62.1 ±1.1 | 6.9 ±2.3 | 7.6 ±1.4 | 42.8 ±1.5 | 22.9 ±1.3 | 38.5 |
| TPR | 0.2 ±0.2 | 98.0 ±3.4 | 65.1 ±30.3 | 65.4 ±18.9 | **0.5 ±0.2** | **0.6 ±0.5** | 40.3 ±3.4 | 26.3 ±7.6 | 37.0 |
| STPR (ours) | **0.0 ±0.1** | **0.1 ±0.2** | **22.8 ±2.3** | **18.9 ±24.1** | 4.3 ±2.9 | 2.3 ±0.0 | **14.4 ±2.8** | **7.9 ±5.0** | **8.8** |

Table 1: Aggregate error on bAbi.

| # examples | 1K | 10K |
|---|---|---|
| MN | 27.3 ± 1.1 | 15.6 ± 0.3 |
| SMN | 25.9 ± 0.4 | 14.4 ± 1.8 |
| TPR | 21.7 ± 2.0 | 2.2 ± 0.9 |
| STPR | 20.3 ± 1.7 | 0.7 ± 0.6 |
| TPR-sm | 17.1 ± 1.0 | 13.0 ± 1.6 |
| STPR-sm | 5.6 ± 0.8 | 3.1 ± 0.4 |
| TPR best | 14.77 | 0.14 |
| TPR-sm best | 13.56 | 7.65 |
| QRN-2r best | 9.9 | 4.6 |
| STPR best | 11.65 | 0.06 |
| STPR-sm best | 1.79 | 1.82 |

Because the embeddings of entities learned by TPR are orthogonal (Schlag & Schmidhuber, 2018), we argue that most dimensions are used to represent this basis (requiring at least as many dimensions as the number of possible entities) while STPR require less dimensions since orthogonal representation are available to the model by construction. To see this effect, we run a smaller model with $d = 20$ and dropout set to 0.5 for both STPR and TPR that are called respectively STPR-sm and TPR-sm. Results indicate that STPR-sm performs almost as well in the 10K setting while reaching an average test error of 5.6% when trained with 1K examples as opposed to TPR-sm whose performance is severely hurt by the dimension reduction as it cannot represent orthogonal basis for the different entities anymore. To the best of our knowledge, the best reported result with 1K examples is 9.9% from QRN model (Seo et al., 2017) who reported the best run over 10 seeds while we report average result over seeds. Using the same procedure, we obtain a test-error of 1.8% and pass 19/20 tasks (passing means reaching an error lower than 5%). This makes a significant improvement to the problem of passing all task with only 1K examples but the problem still stands (in particular when reporting mean test-error instead of best error over multiple seeds).

**Zero-shot entities.** In Tab. 2, we show the accuracy obtained on two bAbi tasks in a zero-shot setting where we leave the training dataset unchanged but perturb the test dataset by introducing unseen entities. Precisely, we replace the 6 rooms present in the task (kitchen, bedroom, office, garden, hallway, bathroom) by kitchenette, guest-room, open-space, entry, terrace, toilet only in the test set, see Fig. 5. We also measure test-accuracy when introducing unseen people and objects in the test set with the same procedure (both tasks contain objects, rooms, and people). For all models, the semantic embeddings of words unseen in the training are set to the zero vector. Symbolic models outperform their semantic-only counter-part in this setting as they can perform abstraction over symbols rather than relying only on specific embeddings for each name.

**Experiment details.** Hyperparameters in bAbi are kept identical across tasks, and we reused hyperparameters of (Sukhbaatar et al., 2015) for Memory-Networks, and Schlag & Schmidhuber (2018) for TPR (they are given in the appendix). For symbolic variants, we use the same hyperparameters as the semantic version. We used the public implementations (Junki, 2015) and (Alexander, 2019) for MN and TPR based on Pytorch (Paszke et al., 2019) that we adapted to support symbolic computation.

## 6    CONCLUSION

We introduced a novel hybrid semantic and symbolic representation able to handle internal transformations and projections of the final representations back to the initial vocabulary in a way that is symbol-shift equivariant. The main advantage of our approach is that we rely on differentiation to detect whether words should behave as symbols and therefore sidestep the need to detect or specify entities in advance.

Our experiments showed that having symbol-shift equivariant models can significantly decrease the amount of training data required. In particular, we were able to solve 19/20 bAbi tasks using only 1K training examples. We also showed that our approach performs well in challenging zero-shot settings or when the number of entities in a task becomes very large.

An interesting area for future work consists in extending other architectures such as Transformer (Vaswani et al., 2017) models with the hope of diminishing the vast amount of data they currently require. Finally, another interesting application will be to use this symbolic representation in order to ensure fairness in question answering or other applications.

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

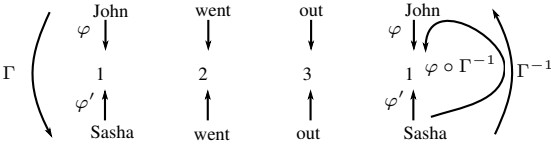

Figure 6: Illustration of proof notations.

## A PROOF OF SYMBOL-SHIFT EQUIVARIANCE

**Theorem.** *SMN and STPR are symbol-shift equivariant.*

*Proof.* Let us take $\Phi \in \{\text{SMN}, \text{STPR}\}$ and show that for any symbol-shift $\Gamma$ and context $X \in \mathcal{X}$, $\Phi(\Gamma(X)) = \Gamma(\Phi(X))$. Since the output of $\Phi$ is considered as a distribution on words, e.g. as a vector $\Phi(X) \in \mathbb{R}^n$, we first define the action of $\Gamma$ on a vector $v \in \mathbb{R}^n$ by defining $v' = \Gamma(v)$ as the vector verifying $v'_{\Gamma(i)} = v_i$ or equivalently $v'_i = v_{\Gamma^{-1}(i)}$ for all $i \leq n$. We then have by definition $[\Gamma(\Phi(X))]_i = [\Phi(X)]_{\Gamma^{-1}(i)}$.

Since $\Gamma$ is a symbol-shift, all word parameters (which comprises $A$, $B$ and $\alpha$) are equivariant, e.g. for each word $i \leq n$:

$$A_{\Gamma(i)} = A_i, \quad B_{\Gamma(i)} = B_i, \quad \alpha_{\Gamma(i)} = \alpha_i \tag{16}$$

Denote $\varphi : C_X \to [1, m]$ and $\varphi' : C_{\Gamma(X)} \to [1, m]$ the two bijections mapping context words in $C_X$ and $C_{\Gamma(X)}$ to their order of appearance, see Fig. 6. Using the fact that permuting words does not change entity order of occurrence, we have:

$$\varphi' = \varphi \circ \Gamma^{-1}. \tag{17}$$

We first show that the embedding defined in Eq. 2 is invariant to symbol-shift, e.g. that $E(x) = E(\Gamma(x))$ for all $x \in X$ where $E(x) = Ae_x \oplus \alpha_x e_{\varphi_X(x)}$. Using Eq. 16, we get that $Ae_x = Ae_{\Gamma(x)}$. Then using Eq. 17 and 16, we get that $\alpha_{\Gamma(x)} e_{\varphi'(\Gamma(x))} = \alpha_{\Gamma(x)} e_{\varphi(x)} = \alpha_x e_{\varphi(x)}$.

Consequently, since embeddings are invariant by symbol-shift and since the model is deterministic, the same latent representation is obtained for both $X$ and $\Gamma(X)$. If we denote the latent function as $h(X) = h_{\text{sem}}(X) \oplus h_{\text{sym}}(X)$ we then have:

$$h(X) = h(\Gamma(X)) \tag{18}$$

Given that the output is defined as

$$\Phi(X) = \beta \text{softmax}\left(B h_{\text{sem}}(X)\right) + (1 - \beta) B_{\varphi X} \text{softmax}\left(h_{\text{sym}}(X)\right),$$

it is sufficient to show that the two functions $f_1(X) = B h_{\text{sem}}(X)$ and $f_2(X) = B_{\varphi X} \text{softmax}\left(h_{\text{sym}}(X)\right)$ are equivariant as softmax and (constant) linear transformations are trivially equivariant. The following sequence of equalities show that $f_1(X)$ is equivariant, it uses $B_i = B_{\Gamma(i)}$ and the fact that the latent function is invariant.

$$[\Gamma(f_1(X))]_i = [\Gamma(B h_{\text{sem}}(X))]_i \tag{19}$$
$$= [B h_{\text{sem}}(X)]_{\Gamma^{-1}(i)} \tag{20}$$
$$= \sum_{k=1}^{d} B_{\Gamma^{-1}(i)k} [h_{\text{sem}}(X)]_k \tag{21}$$
$$= \sum_{k=1}^{d} B_{ik} [h_{\text{sem}}(X)]_k \tag{22}$$
$$= [B h_{\text{sem}}(X)]_i \tag{23}$$
$$= [B h_{\text{sem}}(\Gamma(X))]_i \tag{24}$$
$$= [f_1(\Gamma(X))]_i \tag{25}$$

It remains to show that $f_2(X)$ is equivariant. Denoting the probabilities over symbols $\tilde{p} = \text{softmax}(h_{\text{sym}}(X)) = \text{softmax}(h_{\text{sym}}(\Gamma(X))) \in \mathbb{R}^m$ and using Eq. 5, we obtain

$$[f_2(X)]_i = \begin{cases} \tilde{p}_{\varphi(i)} & \text{if } i \in C_X \\ 0 & \text{else} \end{cases} \quad , \quad [f_2(\Gamma(X))]_i = \begin{cases} \tilde{p}_{\varphi'(i)} & \text{if } i \in C_{\Gamma(X)} \\ 0 & \text{else} \end{cases}$$

Then, for any $i \leq n$,

$$[\Gamma(f_2(X))]_i = [f_2(X)]_{\Gamma^{-1}(i)} \tag{26}$$

$$= \begin{cases} \tilde{p}_{(\varphi(\Gamma^{-1}(i)))} & \text{if } \Gamma^{-1}(i) \in C_X \\ 0 & \text{else} \end{cases} \tag{27}$$

$$= \begin{cases} \tilde{p}_{\varphi'(i)} & \text{if } i \in C_{\Gamma(X)} \\ 0 & \text{else} \end{cases} \tag{28}$$

$$= [f_2(\Gamma(X))]_i \tag{29}$$

$\square$

## B  ARTIFICIAL DATASET EXPERIMENT

To investigate how models generalize with different number of entities, we generate artificial datasets where questions have the form of:

$$x_1 = v_1. \quad \dots \quad x_N = v_N. \quad x_i = ?$$

where we expect $v_i$ as an answer where $v_i$ is the last value assigned to $x_i$ and $x_j \in V_1, v_j \in V_2$ for all $j$. In Fig. 7, we study how well models perform when the number of variables and values increase. Three datasets are generated with $|V_1| = |V_2| = k$ for $k \in \{10, 100, 1000\}$, each dataset has 10K training examples and 1K non-overlapping examples in the test set and each story has $N = 10$ assignments. While the task is elementary, both MN and TPR struggle to generalize to larger number of entities in contrast to their symbolic counterpart SMN and STPR that can solve the task even when with thousands of entities.

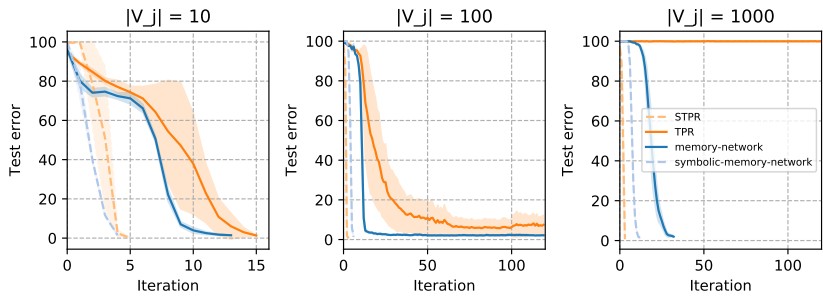

Figure 7: Artificial dataset error convergence when increasing the number of values and variables.

## C  EMBEDDINGS LEARNED BY MEMORY-NETWORKS

Fig. 8 shows the word embeddings of the first layer of a MN after learning the first bAbi task generated with 10k names. The words on the upper-left are verbs, stop-words and locations while the words on the bottom-right are names (the figure is zoomable). The model tries to diagonalize entities embeddings given the available dimensions (e.g. finding orthogonal embeddings for names). This means that attention models may also require a number of dimensions at least as large as the number of entities and hence struggle to generalize to large number of entities or entities not seen at training.

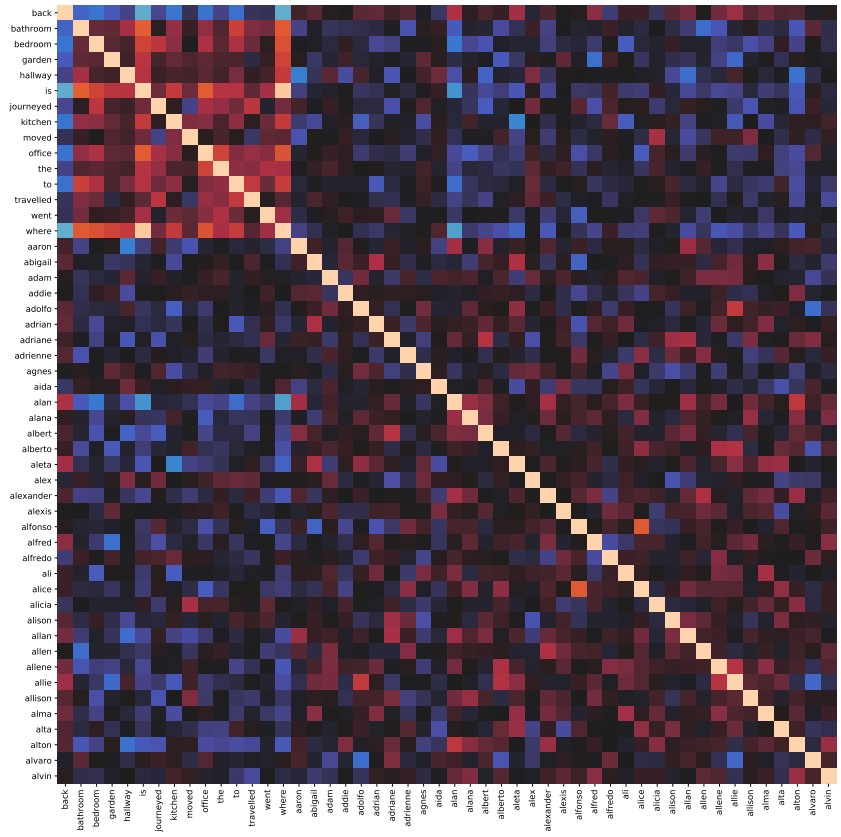

Figure 8: Correlation matrix of word embeddings of the first layer of a MN when trained on first bAbi task generated with 1K names. The learned embeddings of names are approximately orthogonal.

Table 3: Per task error on bAbi with 1K training examples.

| model task | MN | SMN | TPR | TPR-sm | STPR | STPR-sm |
|---|---|---|---|---|---|---|
| 1 | $0.4 \pm 0.3$ | $0.0 \pm 0.1$ | $0.0 \pm 0.0$ | $1.0 \pm 2.2$ | $0.0 \pm 0.0$ | $0.0 \pm 0.0$ |
| 2 | $71.0 \pm 3.1$ | $69.4 \pm 5.9$ | $27.4 \pm 33.7$ | $6.9 \pm 18.3$ | $15.6 \pm 27.4$ | $2.1 \pm 1.3$ |
| 3 | $76.5 \pm 1.2$ | $76.8 \pm 1.0$ | $73.6 \pm 2.6$ | $68.4 \pm 4.3$ | $71.1 \pm 6.5$ | $8.5 \pm 3.8$ |
| 4 | $21.6 \pm 0.7$ | $21.5 \pm 0.4$ | $0.0 \pm 0.1$ | $2.0 \pm 4.2$ | $0.1 \pm 0.1$ | $0.0 \pm 0.0$ |
| 5 | $22.3 \pm 0.9$ | $21.6 \pm 0.8$ | $2.1 \pm 0.5$ | $2.4 \pm 0.6$ | $4.6 \pm 2.9$ | $2.2 \pm 1.9$ |
| 6 | $23.8 \pm 10.8$ | $16.7 \pm 3.3$ | $19.9 \pm 11.5$ | $12.6 \pm 1.9$ | $24.0 \pm 16.4$ | $4.1 \pm 2.8$ |
| 7 | $23.4 \pm 1.6$ | $18.0 \pm 1.0$ | $16.5 \pm 4.7$ | $10.2 \pm 2.8$ | $17.9 \pm 5.2$ | $1.5 \pm 1.8$ |
| 8 | $17.6 \pm 0.8$ | $14.5 \pm 1.3$ | $10.4 \pm 2.7$ | $4.4 \pm 1.7$ | $4.5 \pm 1.6$ | $0.9 \pm 0.6$ |
| 9 | $19.6 \pm 6.7$ | $10.8 \pm 1.3$ | $31.7 \pm 6.8$ | $21.6 \pm 3.6$ | $35.9 \pm 9.3$ | $4.9 \pm 1.2$ |
| 10 | $23.7 \pm 6.4$ | $19.1 \pm 1.9$ | $39.1 \pm 17.4$ | $21.2 \pm 6.9$ | $30.3 \pm 14.2$ | $10.2 \pm 3.3$ |
| 11 | $10.5 \pm 3.3$ | $10.8 \pm 0.9$ | $2.0 \pm 0.8$ | $1.0 \pm 1.4$ | $7.6 \pm 2.8$ | $1.2 \pm 1.2$ |
| 12 | $0.6 \pm 0.2$ | $0.0 \pm 0.0$ | $0.4 \pm 0.2$ | $0.1 \pm 0.1$ | $3.2 \pm 3.1$ | $2.1 \pm 0.8$ |
| 13 | $6.5 \pm 1.1$ | $6.8 \pm 1.3$ | $8.1 \pm 1.7$ | $4.9 \pm 0.6$ | $7.5 \pm 1.8$ | $3.8 \pm 1.6$ |
| 14 | $3.8 \pm 6.6$ | $10.9 \pm 8.7$ | $14.4 \pm 12.6$ | $21.3 \pm 3.8$ | $6.5 \pm 10.3$ | $1.8 \pm 4.3$ |
| 15 | $0.0 \pm 0.1$ | $0.0 \pm 0.0$ | $0.0 \pm 0.0$ | $0.3 \pm 0.3$ | $0.0 \pm 0.0$ | $0.0 \pm 0.1$ |
| 16 | $52.1 \pm 1.4$ | $52.5 \pm 1.4$ | $53.0 \pm 2.1$ | $53.5 \pm 1.0$ | $39.1 \pm 24.5$ | $0.3 \pm 0.4$ |
| 17 | $47.0 \pm 1.6$ | $46.1 \pm 2.2$ | $42.5 \pm 2.9$ | $32.6 \pm 2.2$ | $45.1 \pm 2.0$ | $24.4 \pm 13.0$ |
| 18 | $48.9 \pm 1.6$ | $48.1 \pm 1.9$ | $3.8 \pm 1.3$ | $1.9 \pm 1.1$ | $5.5 \pm 3.0$ | $0.9 \pm 1.2$ |
| 19 | $76.8 \pm 1.1$ | $74.3 \pm 2.8$ | $88.6 \pm 3.7$ | $75.8 \pm 6.8$ | $86.7 \pm 6.4$ | $43.9 \pm 14.2$ |
| 20 | $0.0 \pm 0.1$ | $0.0 \pm 0.0$ | $0.0 \pm 0.0$ | $0.4 \pm 0.4$ | $0.0 \pm 0.0$ | $0.0 \pm 0.0$ |

Table 4: Per task error on bAbi with 10K training examples.

| model task | MN | SMN | TPR | TPR-sm | STPR | STPR-sm |
|---|---|---|---|---|---|---|
| 1 | $0.0 \pm 0.0$ | $0.0 \pm 0.0$ | $0.0 \pm 0.0$ | $0.0 \pm 0.1$ | $0.0 \pm 0.0$ | $0.0 \pm 0.0$ |
| 2 | $23.1 \pm 1.7$ | $25.1 \pm 27.0$ | $0.1 \pm 0.1$ | $0.3 \pm 0.3$ | $0.3 \pm 0.2$ | $0.6 \pm 0.5$ |
| 3 | $29.4 \pm 2.0$ | $23.4 \pm 1.6$ | $1.4 \pm 0.3$ | $25.7 \pm 12.1$ | $1.6 \pm 0.5$ | $6.6 \pm 5.1$ |
| 4 | $22.0 \pm 0.6$ | $21.7 \pm 0.9$ | $0.0 \pm 0.0$ | $6.8 \pm 10.3$ | $4.5 \pm 14.1$ | $0.0 \pm 0.0$ |
| 5 | $13.9 \pm 3.8$ | $6.6 \pm 3.1$ | $0.8 \pm 0.3$ | $0.7 \pm 0.6$ | $0.6 \pm 0.3$ | $1.0 \pm 1.2$ |
| 6 | $3.8 \pm 0.8$ | $7.6 \pm 5.5$ | $0.3 \pm 0.2$ | $15.1 \pm 3.5$ | $0.4 \pm 0.6$ | $1.2 \pm 0.9$ |
| 7 | $2.3 \pm 0.5$ | $6.1 \pm 0.4$ | $0.5 \pm 0.2$ | $6.0 \pm 15.6$ | $0.3 \pm 0.5$ | $0.6 \pm 0.8$ |
| 8 | $3.8 \pm 0.4$ | $2.2 \pm 0.6$ | $0.8 \pm 0.6$ | $3.3 \pm 6.7$ | $0.9 \pm 1.0$ | $0.3 \pm 0.4$ |
| 9 | $2.3 \pm 0.3$ | $2.1 \pm 0.8$ | $0.4 \pm 0.3$ | $10.5 \pm 2.6$ | $0.5 \pm 0.6$ | $1.0 \pm 0.5$ |
| 10 | $5.0 \pm 1.0$ | $3.1 \pm 1.6$ | $0.4 \pm 0.4$ | $19.2 \pm 5.2$ | $0.6 \pm 0.8$ | $3.9 \pm 2.2$ |
| 11 | $0.2 \pm 0.1$ | $0.7 \pm 1.7$ | $0.4 \pm 0.1$ | $0.3 \pm 0.1$ | $0.2 \pm 0.1$ | $0.2 \pm 0.2$ |
| 12 | $0.1 \pm 0.1$ | $0.0 \pm 0.0$ | $0.1 \pm 0.2$ | $6.5 \pm 5.9$ | $0.3 \pm 0.3$ | $1.2 \pm 0.7$ |
| 13 | $1.3 \pm 1.8$ | $5.7 \pm 0.4$ | $0.5 \pm 0.3$ | $4.0 \pm 1.2$ | $0.6 \pm 0.5$ | $0.2 \pm 0.2$ |
| 14 | $0.3 \pm 0.3$ | $0.1 \pm 0.1$ | $0.1 \pm 0.3$ | $37.1 \pm 10.0$ | $0.2 \pm 0.2$ | $1.3 \pm 2.4$ |
| 15 | $0.0 \pm 0.0$ | $0.0 \pm 0.0$ | $0.0 \pm 0.0$ | $0.1 \pm 0.1$ | $0.0 \pm 0.0$ | $0.0 \pm 0.0$ |
| 16 | $51.9 \pm 1.0$ | $31.6 \pm 27.1$ | $0.3 \pm 0.9$ | $52.2 \pm 1.2$ | $0.2 \pm 0.3$ | $0.0 \pm 0.1$ |
| 17 | $42.8 \pm 1.8$ | $41.2 \pm 2.4$ | $1.9 \pm 3.0$ | $14.6 \pm 6.9$ | $0.3 \pm 0.4$ | $1.2 \pm 1.9$ |
| 18 | $45.0 \pm 2.8$ | $42.3 \pm 1.4$ | $0.4 \pm 0.3$ | $0.8 \pm 0.5$ | $0.3 \pm 0.5$ | $0.2 \pm 0.3$ |
| 19 | $65.3 \pm 2.5$ | $69.2 \pm 1.3$ | $35.8 \pm 16.5$ | $57.3 \pm 11.0$ | $1.3 \pm 1.6$ | $42.5 \pm 7.4$ |
| 20 | $0.0 \pm 0.0$ | $0.0 \pm 0.0$ | $0.0 \pm 0.0$ | $0.3 \pm 0.3$ | $0.0 \pm 0.0$ | $0.0 \pm 0.0$ |

## D  EXPERIMENTS DETAILS

In all our experiments, we used a batch-size of 32 and ADAM with a learning-rate of $0.001$. In the case of TPR models, we follow Schlag & Schmidhuber (2018) and set $\beta_1 = 0.6, \beta_2 = 0.4$. Gradients norm are clipped to $5.0$. All models are trained with early-stopping using 10% of the training set as validation, and we perform 80000 gradient updates 2 times decaying the learning rate by 2 each time (which is a setup close to both Sukhbaatar et al. (2015) that decays the learning rate 4 times and Alexander (2019) that trains for significantly longer).

The main difference in the implementation we used for TPR are that RADAM optimizer was used in Schlag & Schmidhuber (2018), we used ADAM instead to have the same setting with Memory-Networks and also because this optimizer is not available in Pytorch. For Memory-Networks, we reuse Junki (2015) implementation where temporal encoding regularization is not present.

**Memory-Networks temporal encoding**  To allow models to preserve temporal information, Sukhbaatar et al. (2015) proposes to have additional parameters $T_A \in \mathbb{R}^{T \times d}$ and $T_C \in \mathbb{R}^{T \times d}$ where $T$ denotes the maximum number of stories and add them to $m_i$, $c_i$ as follows:

$$m_i = \sum_{j=1}^{n_i} A^{(k)} e_{x_{ij}} + T_A(i) \tag{30}$$

$$c_i = \sum_{j=1}^{n_i} A^{(k+1)} e_{x_{ij}} + T_C(i) \tag{31}$$

In the case of Symbolic Memory-Networks, we proceed similarly with an additional symbolic temporal encoding parametrized with $\alpha_A \in \mathbb{R}^T$ and $\alpha_C \in \mathbb{R}^T$ and update $m_i$, $c_i$ as follows:

$$m_i = \sum_{j=1}^{n_i} (A^{(k)} e_{x_{ij}} + T_A(i)) \oplus (\alpha_A(i) \alpha_{x_{ij}}^{(k)} e_{\varphi(x_{ij})}) \tag{32}$$

$$c_i = \sum_{j=1}^{n_i} (A^{(k+1)} e_{x_{ij}} + T_C(i)) \oplus (\alpha_C(i) \alpha_{x_{ij}}^{(k+1)} e_{\varphi(x_{ij})}) \tag{33}$$

In addition both Memory-Networks and Symbolic Memory-Networks use the positional embedding used in TPR.

## E   TPR UPDATE EQUATIONS

After reading each story $s_t$, $F_t$ is updated by adding the sum of three tensors $\Delta F_t = W_t + M_t + B_t$ whose expression is given by:

$$\hat{w}_t = (F_t \bullet_{34} e_t^{(1)}) \bullet_{23} r_t^{(1)} \tag{34}$$

$$W_t = -(e_t^{(1)} \otimes r_t^{(1)} \otimes \hat{w}_t) + (e_t^{(1)} \otimes r_t^{(1)} \otimes e_t^{(2)}) \tag{35}$$

$$\hat{m}_t = (F_t \bullet_{34} e_t^{(1)}) \bullet_{23} r_t^{(2)} \tag{36}$$

$$M_t = -(e_t^{(1)} \otimes r_t^{(2)} \otimes \hat{m}_t) + (e_t^{(1)} \otimes r_t^{(2)} \otimes \hat{w}_t) \tag{37}$$

$$\hat{b}_t = (F_t \bullet_{34} e_t^{(2)}) \bullet_{23} r_t^{(3)} \tag{38}$$

$$B_t = -(e_t^{(2)} \otimes r_t^{(3)} \otimes \hat{b}_t) + (e_t^{(2)} \otimes r_t^{(3)} \otimes e_t^{(1)}) \tag{39}$$

Where $\otimes$ denotes the tensor outer-product and $\bullet_{ij}$ denotes the tensor inner-product.

The term $W_t$ is a write term that allows to encode the association $(e_t^{(1)}, r_t^{(1)}, e_t^{(2)})$ while retrieving the previously assigned entity and removing its previous association. The term $M_t$ allows to associate this removed entity to a different relation $r_t^{(2)}$ (also removing the previously assigned to $r_t^{(2)}$). Finally the term $B_t$ is called a backlink as it allows associative association, we refer to the original paper Schlag & Schmidhuber (2018) for detailed intuitions on the three update terms.

