# OpenReview forum: "Symbol-Shift Equivariant Neural Networks"
_ICLR.cc/2021/Conference — Reject_

### Official Review · AnonReviewer2 · 2020-10-22
**Confusing**

**Rating:** 4
**Confidence:** 2

**Review:**

**Summary**.
This paper proposes a new type of models that are equivariant to entity permutations, which is an important criterion to build language models that can easily generalize to new entities. The authors modified a Memory-Network and a Third-order tensor product RNN to make them symbolic-shit invariant. The new models were evaluated and compared on the 20 bAbi tasks. Results show that the symbolic versions of the models yield better performance than the original ones.

**Positives**.
The topic is of great interest and it is indeed crutial that neural language models become symbol-shift invariant to allow them to better generalize. This work is clearly motivated.

**Confusions**.
The beginning of Section4 mentions that the main idea of this work is to concatenate a regular "semantic" word vector with a "symbolic" representation essentially corresponding to a one-hot vector of the token order of appearance.
In the following paragraphs, the work presented lacks clarity and seems to over-complicate concepts with hard-to-follow math notations. For instance, the “*Mapping words into and from symbolic representations*” paragraph introduces tedious math notations to describes something simple that was clear before, namely, the mapping from tokens to their respective symbolic vector, which is simply defined as the one-hot vector position appearance of this token in the context.
Similarly, the "*Hybrid semantic-symbolic embeddings*" paragraph uses again tedious math notations to describe how semantic and symbolic embedding are concatenated.

Given the confusion presented in Section4, it is currently not clear how adding a one-hot vector to the input embedding can make a neural model symbol-shift equivariant.
In particular, below are the two things I could not understand:
1) The paper mentions that "*all parameters are differentiable*". It is not clear if that also includes the symbolic representation or not? If so, then the initial one-hot vector may not be a one-hot vector after the gradient updates performed during training, which would result in a non-symbolic representation? if it is kept fix during training, then it is not clear how it is used by the network.
2) In addition, assuming that the symbolic representation of all tokens stays the same during training, I don't see how "_permuted symbols share the same latent representations_" if the latent representations are made of both on-hot vectors **and** regular word vectors. I understand that the symbolic representation does not change for a permuted word since it will appear at the same place as the original word. But the semantic representation will be different. For instance, the semantic word vector of “banana” is similar but still different than the word vector of “apple”.

**Conclusion**.
I would suggest the authors to simplify their mathematical notation and make their paper easier to read. As of now, I could not fully understand the paper and unfortunately for that reason could only put a score of 4 with a low confidence of 2.

---

> ### Author Response · Authors · 2020-11-12
> **Answer to Reviewer 2**
>
> Thank you for your review.
>
> To answer your questions:
> 1. The only free-parameter of the symbolic representation $\alpha_x e_\varphi(x)$ is $\alpha_x \in [0, 1]$ given that $e_\varphi(x) \in \mathbb{R}^m$ is set to the one-hot vector of the order of appearance of $x$. With the parameter $\alpha_x$, the model can learn if a word $x$ should behave as a symbol or not.
> 2. In our definition of symbol-shift, permutation of symbols must happen without changing semantic embeddings so that “banana” and “apple” can be changed only if they share the same semantic word vector. This is a theoretical requirement which allows us to prove formally that the models we introduce are equivariant under symbol shift.
> However, in practice as you mention we cannot expect the semantic vector of two entities of the same group to be exactly the same, they would be close but different. Our claim is that the added inductive bias helps in practice, even if the theoretical requirement does not hold, which we aimed to demonstrate in our experiments as the models can generalize to a larger number of entities (for instance in the experiment of Fig 1, the semantic vectors of all entities are close but different), to unseen one and also converges faster than models that do not have the symbolic representation we propose.
>
> About your other points on notations, we will do our best to simplify them and we are grateful for your feedback. We agree that the symbolic vector can be described in simple words, however it is a bit more difficult to express the projection of a symbolic vector to the vocabulary in words without equations.
>
> Please let us know of any other questions you may have.

---

> > ### Comment · AnonReviewer2 · 2020-11-16
> > **Answer to authors**
> >
> > Thank you for taking the time to answer.
> > 1. Thank you for the clarification. This should be better mentioned/explained in the paper.
> > 2. I see that both R1 and R3 had the same confusion. I believe that your explanation is a **crutial** point that the paper should explicitly mention.
> >
> > I agree with the other reviewers that the paper is hard to parse. I think one suggestion would be to simplify the mathematical notations as I suggested in my original comment.
> > for instance, the one-hot vector of the order of appearance of token $x$ could be defined as $e_x$ rather than $e_\phi(x)$. What is $\phi$ anyway? --looking back at the paper-- oh right it is the bijection. yeah so this whole section could be rewritten to avoid confusion. I would suggest to simply talk about matrix $B$ as it is clear with Figure3 that this is just a simple mapping from tokens to 1-hot vector of position. Also, why matrix "$B$" and not "$E$" since the symbolic representation is noted "$e_x$"...?
> > Now that I think about it, matrix $B$ (or $E$ to be consistent with vectors $e_x$) is just an $m \times m$ identity matrix if you organize each row as being the words in order! So yeah really this whole section is very confusing. I think you could just replace it by saying what you said in all your replies to R1, R2, R3, eg:  *the symbolic representation is defined as $\alpha_x e_x$ with parameter $\alpha_x \in [0, 1]$ and $e_x$ being defined as the one-hot vector of the order of appearance of $x$. $\alpha_x$ is trained alongside the other network parameters with gradient descent, while $e_x$ is kept fixed. Note that if $\alpha_x = 0 \forall x$, the model is reduced to a classical "semantic"-only model.*
> >
> > Also, is $\alpha_x$ really $ \in [0, 1]$? I am *guessing* $\alpha_x$ is the output of a sigmoid unit (if it is, you should definitely say it in the paper), so a continuous variable between 0 and 1, and not a binomial variable that you sample for each word, in which case you should actually denote $0< \alpha_x  < 1$ explicitly as the $\in [0,1]$ notation could be confusing and interpreted as being either 0 **or** 1.

---

> > > ### Author Response · Authors · 2020-11-18
> > > **answer**
> > >
> > > Thank you for the useful suggestions.
> > >
> > > To answer to your specific points:
> > >
> > > * we would use your suggestion and refer more to Figure 3 to give the intuition of the projection matrix $B_\varphi$
> > > * we would like to point that $B_\varphi$ is not squared,  $B_\varphi \in \mathbb{R}^{n\times m}$ (its dimension is indicated in "Mapping words into and from symbolic representations")
> > > * we appreciate your suggestion regarding the presentation of the symbolic embedding (we would have to rename it as $e_x$ is already used in the semantic embedding $A e_x$.)
> > > * Yes, $\alpha_x$ is the output of a sigmoid unit, we will add this mention in the paper.

---

> > > ### Author Response · Authors · 2020-11-23
> > > **Updated manuscript based on AnonReviewer2 recommendations**
> > >
> > > We have uploaded a new version of the manuscript with some of your recommendations to enhance the clarity of Section 4

---

### Official Review · AnonReviewer3 · 2020-10-28
**not very well motivated**

**Rating:** 3
**Confidence:** 4

**Review:**

This work proposes to improve the generalizability of bAbi models through [entity permutations].
More specifically the approach assumes domain knowledge of word/entity type equivalences, which helps restricting possible permutations between word POS (e.g., “John” vs “why”) or gender (e.g., “John” vs “Mary”). Each word type has its own embedding param and is concatenated with normal word embeddings to form the final word representation. Experiment with memory networks and Third-order Tensor Product RNN shows that the proposed approach indeed enables the models (especially TPR) to handle artificial data sets with large number of entity names.

Overall I find the proposed research not very well motivated. Leveraging word type knowledge to improve the generalizability of NLP models has been a popular and effective approach. Commonly used strategy is to replace named entities in sentences with their word type tokens . e.g., from [how old is Obama] to [how old is PERSON]
https://arxiv.org/abs/1601.01280
https://arxiv.org/abs/1611.00020
The proposed approach seems to achieve a similar effect, but is a lot more complex.

---

> ### Author Response · Authors · 2020-11-12
> **Clarification on the fact that no domain knowledge is assumed**
>
> Thank you for your review.
>
> There seems to be an important misunderstanding that we would like to clarify. We do not assume that we have “domain knowledge of word/entity type equivalences”. This is in fact the main motivation and contribution of this paper: having compositionality and symbol abstraction **without** having to specify/detect entities in advance.
>
> We hope you can adapt your review as the point you raised is the main motivation and contribution of the paper (as stated in the abstract “we define a class of models whose outputs are equivariant to entity permutations *without requiring to specify or detect entities* in a pre-processing step” or in the introduction “The main advantage and novelty of our approach is that *entities are not required to be identified in advance* as we rely solely on differentiation to determine whether a word acts like an entity”).
>
> In case there is one sentence that is misleading and indicates that we are assuming entities equivalences are given, we would really appreciate it if you could point us to it.
>
> We can add the references you mentioned but we would like to point-out that we already highlighted the fact that replacing entities by token helps compositionality in our related work section (“compositionality becomes much easier if symbols (or entities) are detected before-hand. For instance, [Li2015] showed that replacing entities by dedicated token placeholders leads to significant improvement in question answering.”).

---

> > ### Comment · AnonReviewer3 · 2020-11-15
> > **The algorithm is not clearly described**
> >
> > The description of the method claims that it can restrict possible permutations between word POS (e.g., “John” vs “why”) or gender (e.g., “John” vs “Mary”). It seems impossible without an annotator, which decides the type of words. Imagine a new person name (e.g., Alice) in the test set, which never appears in the training set. How is it possible for the model to tell whether "Alice" is equal to "why" or "John"?
> >
> > The terminology used in this paper also seems non-standard, which make it harder to parse. For example, the so called "symbolic representation" seems to be just the one-hot representation of vocab id.

---

> > > ### Author Response · Authors · 2020-11-15
> > > **answer to comment**
> > >
> > > First of all, thank you for your comment and following-up.
> > >
> > > You are right that if a word is not seen during training, the model cannot infer properties such as gender or POS. We would like to point out that the description of the method does not claim that it can restrict possible permutations. The definition of possible permutations (symbol-shift in the paper) is only required to define the class of symbol-shift equivariant models: e.g. the class of model whose outputs is equivariant to possible permutations of symbols. While the definition is restrictive (having semantic vectors equals for different groups of entities is a stringent condition), it is helpful in our view as it allows to restrict to permutations that does not alter syntax (e.g. preventing permuting "John" by "Who"), it is also helpful as we can prove that some models are indeed equivariant under those conditions.
> > >
> > > The semantic word properties (gender, POS, ...) are learned in a standard way (as noted to R1 if $\alpha_x = 0$ for all words, the model symbolic component vanishes). For instance, if the model sees “John” sufficiently many times, the model learns the word properties with its semantic vector. If the word is never seen during training, our experiments on zero-shot show that the model can still generalize by leveraging the symbolic representation (for words that does not appear in the training, semantic vectors are set to zero).
> > >
> > > Regarding the terminology, we agree that using “symbolic” is not standard but given that we are introducing a new type of equivariant representation, it is unclear to us if there would be a standard terminology (we would appreciate any suggestion). We choose to call "symbolic representation" parts of the model that are equivariant when permuting symbols (e.g. when applying symbol shift). Note that this does not contain *just* the one-hot of the order of appearance in the context but also how this one-hot representation can be projected back to the vocabulary (again preserving the equivariance property) and how this representation can be transformed.

---

> > > > ### Comment · AnonReviewer3 · 2020-11-15
> > > > **full system illustration?**
> > > >
> > > > Using vocab id and token position id for text representation seem pretty standard for deep text models. It might be helpful to have the full system illustrated in Fig 3 with details about how the semantic embeddings (e.g., [1 -1] for John) is generated, and how "restricting"  in the following statement is applied.
> > > >
> > > > "we introduce the notion of symbol- shift equivariance: e.g. a condition restricting possible permutations as some permutations may perturb the sentence grammar (permuting “John” by “why”) or cause ambiguity (permuting “John” by “Mary” if the question involves a gender)."

---

> > > > > ### Author Response · Authors · 2020-11-15
> > > > > **Answer**
> > > > >
> > > > > We agree vocab-id and position-id are standard (in fact the two baselines we consider use vocab-id and position-id). However, here we use the *order of appearance* in the context which is not standard (in particular the way we propose to project such representation to the vocabulary).
> > > > >
> > > > > Note that positional encoding are discussed in related work and in particular the difference with the approach we propose ("While  positional  encoding  (Graves  et  al.,  2014;  Vaswani  et  al.,  2017)  may give some compositionality ...").
> > > > >
> > > > > We will add an illustration of the word embedding (e.g. a matrix with $n \times d$ where the $i$-th row represents John and has coordinates [1, -1]).
> > > > >
> > > > > Regarding, the illustration of restricting valid permutation, we refer to Figure 2 which is illustrating the concept.

---

### Official Review · AnonReviewer1 · 2020-10-30
**Equivariant Networks for NLP**

**Rating:** 5
**Confidence:** 3

**Review:**

The authors propose a network that is equivariant to entity permutations without requiring the pre-specification of the set of entities. To this end, the authors propose a hybrid semantic-symbolic embedding which they integrate into two QA models. Finally, the authors show significant gains on the bAbi tasks, with especially impressive gains in the 1K setting.

The problem is quite interesting and challenging in the setting where entities are not prespecified.
However, given the model description it is not clear at all how the model is able to learn a symbol-shift equivariant embedding.
I don't understand how the model is able to determine that "apple" and "orange" have the same embedding while "apple" and "John" have different embeddings.
What is the loss/model architecture/data augmentation guiding this? How is the model able to figure out that "John" and "Sasha" share embedding?

Apart from the high level details, I don't understand the following notations and operations:
* In Section 4, what is $n$? Is it total number of words in the sequence?
* If $B_\varphi \in R^{m \times n}$  and $e^m_j \in R^m$, the multiplication $B_\varphi e^m_j$ doesn't make sense.
* How exactly is $\alpha_x e_{\varphi(x)} \in R^m$?  What exactly is $\alpha_x$ and what is it's shape?

The notation and the working of the model is not clear to me, hence, I am giving a low rating for now.
Apart from this I also doubt the proposed method's generalizability beyond toy settings.

---

> ### Author Response · Authors · 2020-11-12
> **Answer to Reviewer 1**
>
> Thank you for the review.
>
> “I don't understand how the model is able to determine that "apple" and "orange" have the same embedding while "apple" and "John" have different embeddings.”
>
> The model learns the semantic embeddings in the same way as a standard model (in fact, if $\alpha_x = 0$ for all word $x$, the model will reduce to a “semantic” model). Thus the model is able to learn similar embeddings to “orange” and “apple” that would be close but different (for instance if semantic embedding encodes color) the same way than any word embedding model relying on differentiation. While we assume that vectors of entities in the same group are equal, this is only required to prove formally that models are equivariant. In practice as noted by R2, the semantic vectors are only *close* but not equal between entities of the same group. However, our claim is that this additional inductive bias still allows the model to generalize as seen in our experiments where “symbolic” models are able to learn with large number of entities or unseen ones even if the condition of the theorem does not strictly apply.
>
>  In regards to your specific points:
> * $n$ is the number of words in the vocabulary, it is introduced in Section 3 paragraph 2, we will recall its definition in Section 4 to ease readability.
> * We did a typo when indicating the dimension of  $B_\varphi$ which is $B_\varphi \in R^{n \times m}$
> * $\alpha_x$ is the $x$-th component of $\alpha \in [0, 1]^n$ and hence $\alpha_x \in \mathbb{R}$. $e_{\varphi(x)} \in R^m$ as it is the one-hot vector of the order of appearance of $x$ in the context (which has $m$ words).
>
> We hope this clarifies the points raised, please let us know of other questions you may have.

---

> > ### Comment · AnonReviewer1 · 2020-11-21
> > **Follow up**
> >
> > I think after the clarifications I have a better grasp of the work.
> > It seems like all the reviewers had difficulty parsing the work.
> > I highly recommend following through on some of Reviewer 2's suggestions to improve the writing.
> >
> > There are a few things which are misleading/overclaimed statements in the paper:
> > * From what I understand, the proof of equivariance hinges around the semantic vectors being the same for entities behaving the same way - (John, Sasha), (apple, orange). When one reads the paper, these are not stated as assumptions but rather part of what's obviously true -- See equation 16 in Appendix A, specifically the assumption that   $A_{\tau_{(i)}} = A_i$. Unless it's obvious from context, theorems clearly state the assumptions they are making, typically in their statement itself, and I ask the reviewers to the same.
> > *  $A_{\tau_{(i)}} = A_i$ is a huge assumption. In fact, the model is equivariant without even requiring the symbolic part at all. This more or less never happens in practice, and that's why I don't believe this model can generalize beyond toy settings, such as bAbI.
> > * There's a claim made in the introduction -- "we rely solely on differentiation to determine whether a word acts like an entity". I have not seen any follow up on this claim.
> >
> > Some suggestions on notations and writing:
> > * "which is a discrete set of n words" -> Maybe just say "where |V| = n"
> > * Currently square brackets are used for denoting both a continuos range, and a discrete range. In the "Hybrid semantic-symbolic embedding" paragraph, both $\alpha \in \[0, 1\]^n$ and $x \in \[1, n\]$ are used. May be use $x \in$ {$1, \cdots, n$} and  $\alpha_x \in \[0, 1\]  \forall x \in$ {$1, \cdots, n$}.
> >
> > My current impression of the paper are:
> > * There's a lot of unnecessary mathematical jargon that could be simplified a lot.
> > * The current assumptions made by the proofs are too strong, and the assumptions are not highlighted well enough.
> > * I highly doubt that this formulation can generalize beyond toy settings.

---

> > > ### Author Response · Authors · 2020-11-23
> > > **Updated the Manuscript to improve the writing**
> > >
> > > Thank you for following-up. We updated the manuscript version with feedback from Reviewer 2 and some of your feedback. We agree that there is a clear consensus of the reviewers in some aspect of our presentation that we will work on clarifying.
> > >
> > > * We regret that the assumptions of the theorem were not clear and will make it clearer in the context. While the theorem states that "SMN and STPR are symbol-shift equivariant.", where symbol-shift equivariant definition implies that $A_{\tau_{(i)}} = A_i$, it is clear to us now that this should be better highlighted and explained.
> > >
> > > > "There's a claim made in the introduction -- "we rely solely on differentiation to determine whether a word acts like an entity". I have not seen any follow up on this claim. "
> > >
> > > This refers to the parameter $\alpha_x$ that is learned per word, for instance in Section 4: "where $\alpha_x$ is ... a learnable parameter, $0< \alpha_x<1 $, that indicates how much each word should behave as a symbol".
> > >
> > > > "$A_{\tau_{(i)}} = A_i$ is a huge assumption. In fact, the model is equivariant without even requiring the symbolic part at all.".
> > >
> > > Regarding the assumption, we believe the problem comes mainly from our presentation and we will make this point clearer. The assumption $A_{\tau_{(i)}} = A_i$ is not needed for our model (as shown in the experiments where this property obviously does not hold), it is needed only in our proof and symbol-shift definition.
> > >
> > > The reason why we use this strong assumption is to be able to restrict the class of possible word permutations without knowing entities groups: clearly, there must be some restriction as we cannot expect the meaning of sentences to be preserved if we allow permuting "John" and "the" for instance. The classic definition allows permutation only between words of the same group but this imposes knowing the group while our definition of symbol-shift avoids this need.
> > >
> > > Regarding the fact that a model with $A_{\tau_{(i)}} = A_i$ with no symbolic will be equivariant: this is true but such a model would not be able to distinguish entities in the same group.
> > >
> > > To summarize, the assumption $A_{\tau_{(i)}} = A_i$ is only required to investigate theoretical properties of the model. We will add experiments with another dataset to show that the approach proposed can also work out of bAbI in real-world settings.

---

### Author Response · Authors · 2020-11-12
**General answer to reviewers.**

We would like to thank all the reviewers for their work and for highlighting the merits of our paper in particular:
- having compositionality without specifying entities in advance in Neural Network which is a well motivated and a challenging problem (R1, R2)
- Significant performance gains of the symbolic models and especially in the 1K setting  (R1,R2)

We would like to point out that the criticism of R3 (that we need to detect entities in advance which is not novel) is a misunderstanding. We do not assume entities or any domain knowledge to be specified in advance. This is in fact the main contribution of our approach: to provide some compositionality without requiring such information.

With respect to proposed enhancements, two reviewers ask for model clarifications that we answered in our comments, we will update our manuscript in the next days to reflect those points. In addition, we share the code privately (that we intend to release upon publication) in the hope it can also bring some clarification.

---

### Decision · Program_Chairs · 2021-01-07
**Final Decision**

**Decision:**

Reject

**Comment:**

This paper proposed a  new type of models that are invariant to entities by exploring the symbolic property of entities. This problem is important in language modeling since it gives intrinsically more proper representation of sentences, which can better generalize to new entities.  However I still suggest to reject this paper for the following reasons
1. The description of model is not clear enough which can certainly use a serious round of revision.
2. The experiments on bAbi is not convincing enough since it is an overly simple and toyish data-set with many ways to hack
3. Similar entity-invariant idea has been explored long time ago by  (https://arxiv.org/pdf/1508.05508.pdf) which attempted to represent entities as “variables”